# Progression-Free Survival and Treatment-Free Interval in Head and Neck Cancer with Long-Term Response to Nivolumab: Timing of Active Discontinuation

**DOI:** 10.3390/cancers16142527

**Published:** 2024-07-12

**Authors:** Mioko Matsuo, Muneyuki Masuda, Moriyasu Yamauchi, Kazuki Hashimoto, Ryunosuke Kogo, Masanobu Sato, Shogo Masuda, Takashi Nakagawa

**Affiliations:** 1Department of Otorhinolaryngology, Graduate School of Medical Sciences, Kyushu University, 3-1-1 Maidashi, Higashi-ku, Fukuoka 812-8582, Japan; hashimoto.kazuki.416@m.kyushu-u.ac.jp (K.H.); ryukogo@gmail.com (R.K.); tjnrq528@yahoo.co.jp (M.S.); shogo.masuda117@gmail.com (S.M.); nakagawa.takashi.284@m.kyushu-u.ac.jp (T.N.); 2Department of Head and Neck Surgery, National Hospital Organization Kyushu Cancer Center, 3-1-1 Notame, Minami-ku, Fukuoka 811-1395, Japan; mmuneyuki@icloud.com; 3Department of Otolaryngology, Head and Neck Surgery, Faculty of Medicine, Saga University, 5-1-1 Nabeshima, Saga 849-8501, Japan; yamauchimoriyasu@gmail.com

**Keywords:** recurrent/metastatic head and neck squamous cell carcinoma, nivolumab, long-term responder, active treatment discontinuation, timing

## Abstract

**Simple Summary:**

This retrospective study aimed to determine the optimal timing of active discontinuation of long-term responders to immune checkpoint inhibitors in patients with recurrent/metastatic head and neck squamous cell carcinoma. We analyzed treatment duration and treatment-free interval (TFI) in 227 nivolumab-treated patients and determined the timing when progression-free survival (PFS) leveled off and when patients discontinued for unplanned reasons (toxicity or patient decision). The 3-year and 6-year PFS was 15.9% and 15.3%, respectively. The PFS curve was completely flat at 3 years. The median time for patients to request discontinuation was 36.8 months, with a median TFI of 15.1 months. The median time for discontinuation due to toxicity was 18.9 months, with a median TFI of 30.6 months. Given that the PFS curve completely leveled off at 3 years and the median time of discontinuation at the patient’s choice was 3 years, we suggest considering treatment completion at 3 years.

**Abstract:**

The optimal timing for actively discontinuing immune checkpoint inhibitor therapy in long-term responders with recurrent/metastatic head and neck squamous cell carcinoma (R/M HNSCC) remains unresolved. We conducted a retrospective study of 246 patients with R/M HNSCC treated with nivolumab to determine the optimal timing to actively discontinue nivolumab therapy. We examined the point at which progression-free survival (PFS) plateaued in all cases. We compared the prognosis of 19 (7.7%) ongoing cases and 227 (92.3%) discontinued cases and analyzed treatment duration and treatment-free interval (TFI). The 6-year overall survival was 11.8% (median, 12.1), and the 6-year PFS was 15.3% (median, 3.0). The PFS curve remained stable for 3 years. The median duration of nivolumab treatment was 2.9 months (range 0.03–81.9): Ongoing group, 41.8 (5.6–81.9); Decision group, 36.8 (4.0–70.1); Toxicity group, 30.6 (2.8–64.8); and progressive disease group, 2.0 (0.03–42.9). TFI in the Decision group was 15.1 months (0.6–61.6) and 30.6 months (2.8–64.8) in the Toxicity group. Long-term responses in R/M HNSCC patients treated with nivolumab are rare but gradually increasing. For this patient group, our best estimate of the optimal time to end treatment is 3 years, as the PFS in this study reached a plateau at that timepoint.

## 1. Introduction

Recurrent/metastatic head and neck squamous cell carcinoma (R/M HNSCC) has a poor prognosis of 3–6 months if untreated [1,2]. Although cytotoxic anticancer drugs and molecularly targeted agents can prolong life expectancy by several months, long-term survival is not expected [3]. However, immune checkpoint inhibitors (ICIs) are capable of producing long-term survival [4,5]. After the efficacy of nivolumab in the CheckMate-141 trial [4] and pembrolizumab in the KEYNOTE-048 trial [5] was reported for R/M HNSCC, ICI was rapidly introduced into clinical practice and is now one of the first choices of drug therapy [6]. There are multiple reasons for this. The main reasons for its use in clinical practice are that it was as effective as or more effective than cytotoxic anticancer drugs and molecularly targeted agents [4,5] and that it had a low frequency of adverse events that could impair the quality of life of patients [4,5]. Moreover, it has since been confirmed that there are rare cases of long-term effects in actual clinical practice [7,8], which is an important factor in the choice of anti-cancer drug therapy. Patients with R/M HNSCC who have undergone ICI treatment have a 2-year overall survival (OS) of 16.9% [7] and a 5-year OS of 19.2% [8]. However, very few patients have achieved such a benefit [6,7], and due to this, there remains no consensus on whether ICI therapy should be discontinued in such patients and, if so, the appropriate time to discontinue [9]. Long-term response to ICI is observed in patients with various types of cancers [10,11], and the timing of ICI discontinuation has been discussed in previous studies [10,11]. Although discontinuation times in long-term responders have been suggested, such as 2 years in lung cancer [10] and 6 months to 2 years in malignant melanoma [11], this debate remains unresolved in other cancers.

Disease progression can be controlled by continuing treatment, but with the risk of treatment-related toxicity, subsequent loss of quality of life [12], and healthcare economic issues [13]. In contrast, discontinuation of treatment may be accompanied by disease progression. Both scenarios are distressing for both patients and clinicians. These problems are not limited to ICI alone. The introduction of anti-cancer drugs, mainly molecular-targeted agents, into clinical practice has increased markedly over the past few years [14], making it possible to switch to the next drug even when the anti-tumor effect of the current one is no longer available and allowing for multiple drug changes. It is very good to have the next means of tumor control. However, we need to consider the harmful influence of long-term continuous administration of anti-cancer drugs. We believe that the establishment of sustainable drug therapy, including treatment-free periods, is an important issue for the future. ICI has been employed in clinical practice for more than 7 years; thus, it is time to conclude the timing of active discontinuation in long-term responders. The purpose of this study was to clarify the timing of active discontinuation in long-term responders to nivolumab in R/M HNSCC by examining the following three points: (1) whether the progression-free survival (PFS) curve plateaus and, if so, when; (2) whether long-term nivolumab treatment increases the frequency of immune-related adverse events (irAEs); and (3) what are the prognoses, duration of treatment, and treatment-free intervals (TFIs) of patients who discontinued treatment for reasons other than the physician’s ability to actively declare the end of treatment (irAE or their own reasons). The aim of this study is to use these data to suggest the optimal timing to actively discontinue nivolumab therapy.

## 2. Materials and Methods

### 2.1. Patients

We conducted a multicenter, retrospective study at Kyushu University Hospital (Fukuoka, Japan) and affiliated institutions. Our study included R/M HNSCC cases treated with nivolumab between April 2017 and December 2023. The observation period was defined as until death or a cut-off date (March 2024), and the median observation period was 11.5 months (range 0.3–82.9). Nivolumab was administered on a schedule of 240 mg/body weight once every 2 weeks or 480 mg/body weight once every 4 weeks. Antitumor effects were evaluated on computed tomography at intervals of 8–12 weeks. Nivolumab was continued until progressive disease (PD) occurred, severe irAE incidence, or patients’ decision to discontinue. The decision to discontinue ICI owing to the incidence of irAEs was made according to each center’s regulations, which were based on the same criteria. Severe irAEs that precluded re-institution of ICI were interstitial pneumonia, neuropathy, encephalitis, myocarditis (any grade), and hepatic and renal dysfunction (grade ≥ 2 and unresponsive to prednisone therapy). Patients who developed irAEs had a period of temporary withdrawal but resumed treatment.

The study was approved by the Institutional Review Board of Kyushu University (IRB No 22011-01) and each participating institution. All patients provided consent to participate in this study. The procedures followed were in accordance with the principles of the Declaration of Helsinki.

### 2.2. Definitions

PFS was defined as the interval from nivolumab administration until PD, and OS was defined as the interval from nivolumab administration until death. The Decision group comprised patients who voluntarily discontinued treatment, the Toxicity group comprised patients who had to discontinue treatment owing to severe irAEs, and the PD group comprised patients who exhibited PD almost simultaneously with discontinuation of treatment owing to irAEs. Treatment-free interval (TFI) was defined as the interval from the last nivolumab administration until the date of confirmed disease-free survival.

Combined Positive Score (CPS) was calculated as the number of programmed death–ligand 1 (PD-L1) positive cells (tumor cells, macrophages, and lymphocytes)/total number of tumor cells × 100 using the 22C3 PharmDx assay (Dako platform, Agilent Technologies, Santa Clara, CA, USA).

### 2.3. Statistical Analysis

Statistical Package for Social Sciences version 22.0 (IBM Japan, Ltd., Tokyo, Japan) was used for statistical analysis. OS and PFS were estimated using the Kaplan–Meier method. Differences between groups were calculated using the log-rank test with Fisher’s exact test for categorical variables and the Kruskal–Wallis test for continuous variables. A *p*-value < 0.05 indicated statistical significance.

## 3. Results

### 3.1. Baseline Patient Characteristics

At the March 2024 timepoint, 19 (7.7%) patients were in the Ongoing group and 227 (92.3%) in the discontinuation group (9 [3.7%], Decision; 12 [4.9%] Toxicity; and 206 [83.7%] PD group). Patient characteristics are described in Table 1. No significant differences were observed between the groups in sex, age, performance status (PS), primary site, platinum sensitivity/resistance, or PD-L1 expression.

### 3.2. Clinical Outcomes Analysis

Overall, the 6-year OS was 11.8% (median, 12.1 months), and the 6-year PFS was 15.3% (median, 3.0 months) (Figure 1). PFS was 22.6% at 1 year, 17.8% at 2 years, 15.9% at 3 years, and 15.3% after 4 years; PFS remained unchanged at 3 years. The 6-year OS of the Ongoing, Decision, and PD groups was 100%, 45.7%, and 2.9%, respectively. The longest observation period in the Toxicity group was 5 years and 6 months, and the OS was 75.0%. Only the PD group showed a significantly worse prognosis (PD vs. Ongoing: *p* = 0.000, PD vs. Decision: *p* = 0.000, PD vs. Toxicity: *p* = 0.000). However, no significant difference was observed between the Ongoing, Decision, and Toxicity groups (Ongoing vs. Decision: *p* = 0.144, Ongoing vs. Toxicity: *p* = 0.264, Decision vs. Toxicity: *p* = 0.675; Figure 2).

### 3.3. Frequency of irAEs

Overall, the incidence of irAEs in the Ongoing group was 42.1% (8/19), in the Decision, Toxicity, and PD groups, these rates were 44.4% (4/9), 100% (12/12), and 23.8% (49/206), respectively. The probability of new irAEs occurring after 2 years was 0.0% (0/19) in the Ongoing group, 11.1% (1/9) in the Decision group, 8.3% (1/12) in the Toxicity group, and 0.5% (1/206) PD group (Table 2).

### 3.4. Duration of Nivolumab Treatment and TFI

Overall, the median duration of nivolumab treatment was 2.9 months (range 0.03–81.9): Ongoing, 41.8 months (5.6–81.9); Decision, 36.8 months (4.0–70.1); Toxicity, 18.9 months (0.03–50.3); and PD, 2.0 months (0.03–42.9).

TFI in the Decision group was 15.1 months (0.6–61.6) and 30.6 months (2.8–64.8) in the Toxicity group. For 52.9% (109/206) of PD cases, second-line treatments such as paclitaxel, cetuximab, or TS-1 were administered, and 47.1% (97/206) were switched to best supportive care (Table 2).

## 4. Discussion

To our knowledge, this is the first report proposing a timeframe for active discontinuation based on TFI and PFS for R/MHNSCC patients treated with nivolumab.

The introduction of ICIs, which are human IgG4 monoclonal antibodies, including nivolumab, has brought dramatic changes to R/M HNSCC treatment [15,16]. In the past, patients who could not be treated with surgery or radiation therapy were treated with cytotoxic anticancer agents or molecularly targeted agents. However, the duration of efficacy was mostly in months rather than years [3]. As such, clinicians faced the challenge of achieving and maintaining antitumor effects. ICI treatment results in long-term sustained complete response (CR) and is continued on a yearly basis, although for a small percentage of R/M HNSCC patients [7,8]. Several studies have already shown that patients with R/MHNSCC can benefit from ICI and be long-term responders [4,5,17,18]. Predictors of efficacy known before ICI administration include PDL-1 [4], CPS [5], PS [17], and Inflammation-based Prognostic Score (IBPS) [17,18], including neutrophil-to-lymphocyte ratio (NLR). Recently, albumin levels have been shown to be a predictor of long-term responders [19]. In addition, it has been reported in various carcinomas that the case with CR is a factor that can be a long-term responder identified after ICI administration [20,21,22,23,24]. Similarly, cases that develop irAEs are also considered to have a good prognosis, which is also a well-known fact in many carcinomas [25,26]. Although only a small percentage of patients with R/MHNSCC achieve such a sustained effect, the number of such cases with the difference between the time ICI became available and the present is accumulating in clinical practice. Therefore, it has become necessary to examine the appropriate timing of ICI discontinuation in cases of persistent CR. Currently, the widely accepted approach in R/M HNSCC is to administer ICI until disease progression or unacceptable toxicity such as irAEs [9]. No guidelines exist regarding the timing of ICI discontinuation in persistent CR cases. Reportedly, patients have sustained responses for up to 59.2 months [19], and no restrictions have been implemented on the long-term use of nivolumab in real-world clinical practice. The decision to discontinue in long-term responder cases, except in cases of adverse events, is currently at the discretion of patients.

It is necessary to examine this decision from four perspectives to offer discontinuation to long-term ICI responders with maintained CR. The first is the determination of CR. The head and neck region has inherently complex anatomy, which is further complicated by changes and modifications in the hard and soft tissues after surgical or radiation therapy [27]. The use of the Neck Imaging Reporting and Data System (NI-RADS), which allows for standardized and appropriate reporting [27], and the combined use of PET [6] have been reported as useful tools to determine CR despite difficulties in reading and interpretation. In addition, we believe that the first point in deciding whether to abort is to discuss the CR decision in a setting where the surgeon, the radiation therapist, and the radiologist can share the same point of view and images. Second, irAEs are known to occur initially and after several months to a year [28]. Thus, if irAEs increase with long-term administration, ICI treatment should be discontinued. However, a study of irAEs after ICI treatment in 50,347 patients reported that 98–99% of irAEs appeared within 18 months, with no subsequent increase in frequency [28]. The same was true in our study, with only 1.2% of cases developing irAEs after 2 years. Hence, as the frequency of irAEs does not increase over time, it did not seem reasonable to limit the duration of continuous administration because of the risk of developing irAEs.

Third, the relapse rate and time to relapse after discontinuation. In lung cancer patients treated with ICI for >18 months, treatment was discontinued in 50% of cases (including 22% who discontinued owing to toxicity), and progression was subsequently observed in 33% of patients with a median time to progression of 10.0 months [29]. In malignant melanoma, the progression-free survival rate was 78.4% after an additional 2 years of follow-up for patients who completed 2 years of ICI treatment [24]. In a report involving several types of solid tumors, 15% of patients discontinued ICI (approximately 5% owing to CR and 3% owing to irAEs); 12% of those who discontinued owing to CR relapsed [29]. Therefore, even in patients with stable disease after long-term treatment, 10–30% of patients relapse after discontinuation. At this stage, discontinuation of treatment in patients who have achieved CR or after long-term ICI treatment must be carefully and individually considered, whether in malignant melanoma [24] or lung cancer [30]. On the other hand, in ICI treatment for lung cancer, the median duration of treatment was approximately 2 months. The median TFI was approximately 4 months for all patients, but when the number of patients who completed 2 years of ICI was limited, the median TFI was approximately 3 years. More than half of the patients were still without treatment after 3 years. Therefore, completing ICI in 2 years is reported to be reasonable [10]. To date, we have not systematically discontinued nivolumab in any of our patients. However, in daily practice, TFIs occasionally occur owing to adverse events or unplanned personal circumstances. We found that the TFI for patients who discontinued treatment owing to irAEs was prolonged for a median of 30 months, confirming that TFI is effective in suppressing disease progression in the long term after ICI discontinuation, even in a small percentage of patients with severe irAEs. The Decision group also achieved a median TFI of 15 months, confirming the low relapse rate and efficacy in maintaining CR after discontinuation of ICI in R/M HNSCC. Inferring from the favorable results of unplanned TFI, we believe it would be feasible to proactively consider planned discontinuation for long-term ICI responders in R/MHNSCC. Dramatic advances in cancer drug therapy have increased the number of cases with long-term survival. In these modern times, not only aspects of tumor control but also aspects of the patient experience, such as QOL, are becoming more important. The TFI interval is synonymous with relief from the patient’s treatment burden, which seems to correlate with QOL. Therefore, we believe that the study with the TFI perspective is important in establishing sustainable anti-cancer drug therapy from the care providers’ and patients’ viewpoints.

Fourth, the optimal time for planned discontinuation. The duration of ICI administration is associated with the relapse rate after treatment discontinuation. In a study of solid tumor cases, discontinuation of ICI treatment within 12 months was associated with a higher risk of recurrence than discontinuation after 12 months [31]. Similarly, in lung cancer patients treated with pembrolizumab, the median PFS of 24.7 vs. 9.4 months was better in the continuous treatment group than in the group that discontinued treatment at 1 year; discontinuation at 1 year should be carefully considered [32]. Additionally, long-term data from lung cancer patients also reported that 2 years of continuous ICI treatment significantly reduced the risk of disease progression [33]. Other lung cancer reports also suggest that an optimal ICI discontinuation period should be considered because a certain percentage of patients relapse even when the duration of treatment is extended 1–2 years, and deaths due to recurrence occur even after long-term ICI treatment [34]. Six months of treatment is sufficient for patients with CR in malignant melanoma; however, in other cases, treatment should be continued for at least 2 years and possibly indefinitely [11]. In our study, the appropriate time of discontinuation was considered when the PFS curve plateaued. The PFS curve almost plateaued 2 years after the start of treatment and plateaued completely at 3 years, which we considered to be the recommended timing for discontinuation. Additionally, since the median time in the Decision group was 36.8 months, we believe that 3 years coincides with a period when patients can achieve a certain level of reassurance. Therefore, we propose that completing treatment in 3 years is the most likely timeframe when physicians and patients can reach an agreement.

A limitation of this study is the small sample size of patients in the Decision group. Additionally, the median TFI of the Decision group patients was only 15 months; therefore, we could verify whether re-increase does not occur even in the long term. Further studies, including follow-up of these patients, are warranted to resolve this limitation. Moreover, we plan to investigate this in the future. In addition, based on the results of this study, we believe that physicians will actively propose planned discontinuation of ICI in the future. This will help accumulate more evidence of active ICI discontinuation and clarify the validity of the timing of ICI discontinuation, which is our future study plan.

## 5. Conclusions

The prognostic impact of ICI discontinuation in R/M HNSCC patients treated with ICI who progressed to long-term responders is unknown. We found a few patients, albeit a limited number, who had unplanned discontinuation of nivolumab and gained a TFI for >1 year. Regarding the minimum optimal duration of treatment that would not be considered overtreatment for this subset of patients, the data show that PFS completely plateaued at 3 years, the median time patients wished to discontinue treatment was approximately 3 years, and the incidence of irAEs reduced after 2 years. Considering these three perspectives, we propose completion of treatment in 3 years. We believe this proposal is an important step in establishing sustainable anti-cancer drug therapy from both the provider and patient sides.

## Figures and Tables

**Figure 1 cancers-16-02527-f001:**
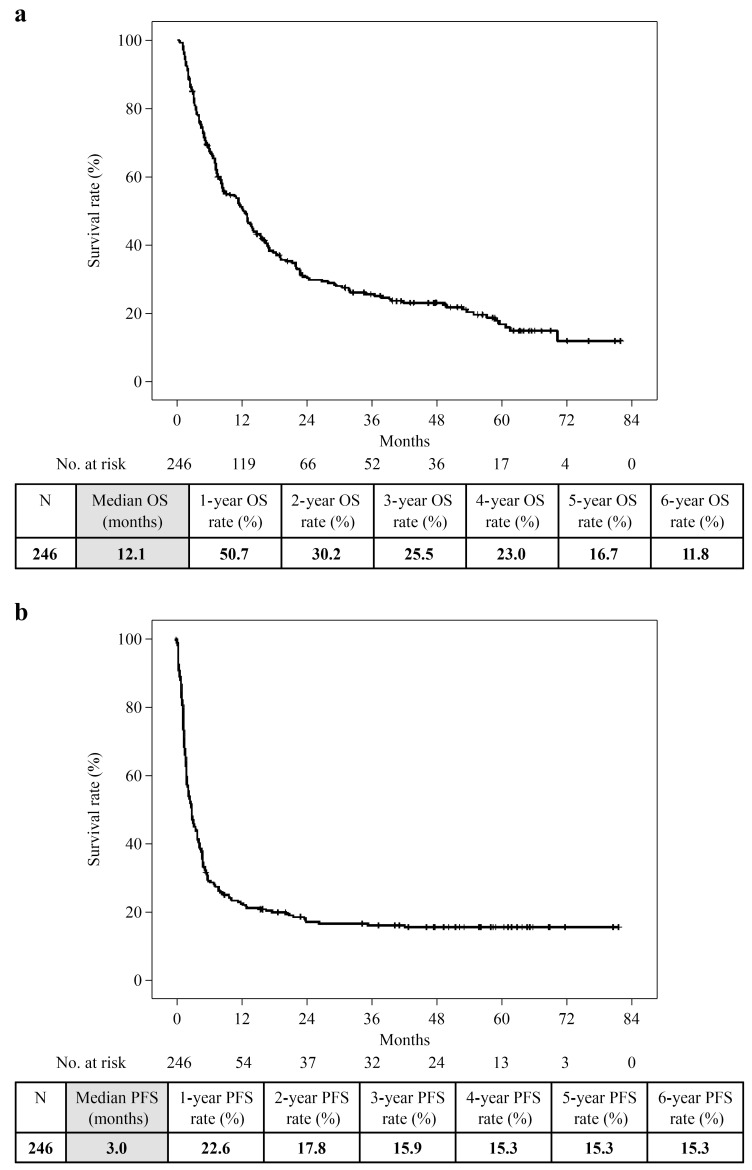
(**a**) Overall survival curves for 246 patients; (**b**) Progression-free survival curves for 246 patients.

**Figure 2 cancers-16-02527-f002:**
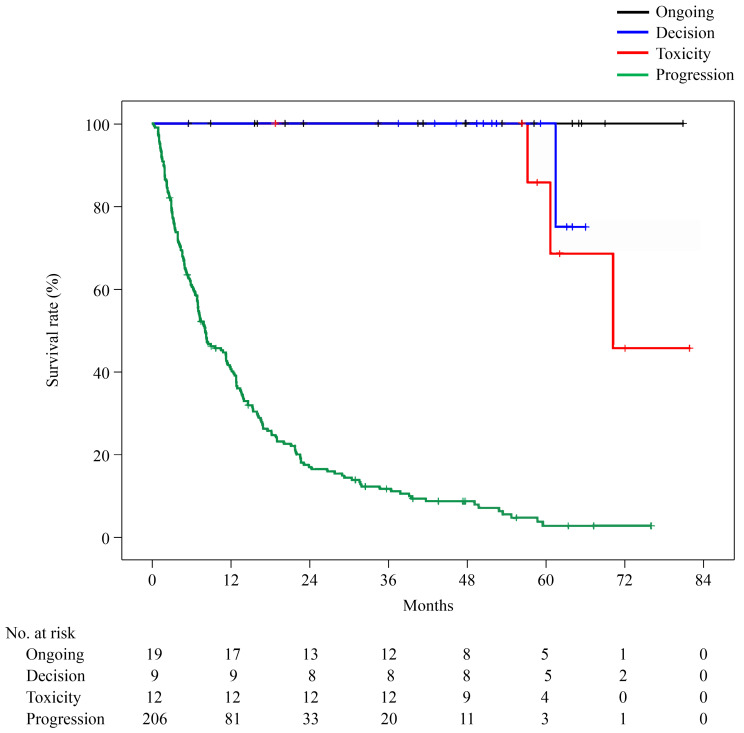
Overall survival curves for the Ongoing, Decision, Toxicity, and PD groups. Only the PD group had a significantly worse prognosis; no significant difference was observed between the other three groups (Ongoing/Decision/Toxicity).

**Table 1 cancers-16-02527-t001:** Patients’ characteristics.

	Total (N = 246)	Ongoing (N = 19)	Discontinuation			*p* Value
			Decision (N = 9)	Toxicity (N = 12)	Progression (N = 206)	
Sex						
Male Female	195 51	14 5	5 4	11 1	165 41	0.193 ^1^
Age						
<75 years ≥75 years	206 40	14 5	8 1	9 3	175 31	0.393 ^1^
Median (range)	66 (23–87)	61 (23–78)	65 (50–80)	63 (54–82)	66 (24–87)	0.601 ^2^
PS						
0–1 2–4	209 37	18 1	9 0	12 0	170 36	0.161 ^1^
Primary site						
Nasopharynx Oropharynx Hypopharynx Larynx Oral cavity Sinonasal tract External auditory canal Salivary gland Primary unknown	9 40 65 17 70 30 10 2 3	0 3 5 1 3 4 1 0 2	1 1 2 0 2 3 0 0 0	0 2 5 2 3 0 0 0 0	8 34 53 14 62 23 9 2 1	0.191 ^1^
Platinum						
Sensitive Resistant	57 189	4 15	4 5	2 10	47 159	0.479 ^1^
PD-L1						
Positive 1–20 ≥21 Negative Not measured	29 42 10 165	3 6 0 10	0 2 1 6	0 0 0 12	26 34 9 137	0.316 ^1^

PD-L1, programmed death–ligand 1; PS, performance status. *p* value determined using ^1^ Fisher’s exact test and ^2^ Kruskal–Wallis test.

**Table 2 cancers-16-02527-t002:** Duration of nivolumab treatment and treatment-free interval.

	Total (N = 246)	Ongoing (N = 19)	Discontinuation		
			Decision (N = 9)	Toxicity (N = 12)	Progression (N = 206)
Duration of nivolumab treatment (median), months	0.03–81.9 (2.9)	5.6–81.9 (41.8)	4.0–70.1 (36.8)	0.03–50.3 (18.9)	0.03–42.9 (2.0)
Treatment-free interval (median), months	—	—	0.6–61.6 (15.1)	2.8–64.8 (30.6)	Second-line treatment or best supportive care
irAE incidence irAE incidence over 2 years	29.7% (73/246) 1.2% (3/246)	42.1% (8/19) 0.0% (0/19)	44.4% (4/9) 11.1% (1/9)	100% (12/12) 8.3% (1/12)	23.8% (49/206) 0.5% (1/206)

## Data Availability

The data sets analyzed in this study cannot be openly shared to protect the privacy of the study participants but are available from the corresponding author upon reasonable request.

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
