# Peer review of "Progression-Free Survival and Treatment-Free Interval in Head and Neck Cancer with Long-Term Response to Nivolumab: Timing of Active Discontinuation"

_cancers, 2024, doi:10.3390/cancers16142527_

Round 1

Reviewer 1 Report

Comments and Suggestions for Authors

This is a very good paper addressing an increasingly important issue. There are just a few minor things that might be addressed.

Abstract lines 36-37: this is not clear and needs rewording.

Figure 1b: the figure legend is labelled Median OS etc, whereas I think you mean Median DFS etc?

Figure 2 legend: "Survival curves" would be better labelled "Overall survival curves" to avoid any confusion.

Frequency of irAEs lines 137-142. All the data is clearly laid out in Table 2 but is quite difficult to follow in the text. I would suggest (ideally) jut summarising the data in the text, or (alternatively) splitting the text to cover separately incidence of irAEs and probability of irAEs beyond 2 years.

You have discussed the dilemma confronting patients and clinicians well in the Discussion. 

Author Response

Response to Reviewer 1

This is a very good paper addressing an increasingly important issue. There are just a few minor things that might be addressed.

Response: I appreciate that assessment from you. I have made corrections. Thank you in advance.

Abstract lines 36-37: this is not clear and needs rewording.

Response: I have changed the sentence. Long-term responses in R/M HNSCC patients treated with nivolumab are rare but gradually increasing. For this patient group, our best estimate of the optimal time to end treatment is 3 years, as the PFS in this study reached a plateau at 3 years.

Figure 1b: the figure legend is labelled Median OS etc, whereas I think you mean Median DFS etc?

Response: You are correct.  I have changed it to PFS.

Figure 2 legend: "Survival curves" would be better labelled "Overall survival curves" to avoid any confusion.

Response:  Thank you very much. I have revised it following your instructions.

Frequency of irAEs lines 137-142. All the data is clearly laid out in Table 2 but is quite difficult to follow in the text. I would suggest (ideally) jut summarising the data in the text, or (alternatively) splitting the text to cover separately incidence of irAEs and probability of irAEs beyond 2 years.

Response: I have changed the sentence. The incidence of irAEs in the Ongoing group was 42.1% (8/19), in the Decision, Toxicity, and PD groups, these rates were 44.4% (4/9), 100% (12/12, and 23.8% (49/206), respectively. The probability of new irAEs occurring after 2 years was 0.0% (0/19) in the Ongoing group, and 11.1% (1/9) in the Decision groups, and 8.3% (1/12) in the Toxicity groups, and and 0.5% (1/206) PD groups.

You have discussed the dilemma confronting patients and clinicians well in the Discussion.

Response: Thank you very much.

Reviewer 2 Report

Comments and Suggestions for Authors

This article aims to determine the optimal timing of discontinuation of immune checkpoint inhibitors in patients of head and neck squamous cell carcinoma. The aim of this study was very important. Several points should be considered.

1.       Presented results were small. Additional analysis about factors, which had a possibility affect the results might be considered, for example, age, smoking history, stage, other therapies including chemotherapy and/or irradiation.

2.       Was PD-L1 score affected the results?

3.       Was there any difference between recurrent tumor and primary carcinoma? 

4.       Was the primary site (nasopharynx, oropharynx and so on) not affect the results?

Author Response

Response to Reviewer 2

 This article aims to determine the optimal timing of discontinuation of immune checkpoint inhibitors in patients of head and neck squamous cell carcinoma. The aim of this study was very important. Several points should be considered.

Response: Thanks for pointing that out. I have made corrections accordingly.

  1. Presented results were small. Additional analysis about factors, which had a possibility affect the results might be considered, for example, age, smoking history, stage, other therapies including chemotherapy and/or irradiation.

Response: You are correct. However, in this case, we are looking at patients with relapsed disease who had to be treated with nivolumab. For this reason, we considered the stage at the time of initial treatment to be unimportant. If we were to assign a stage at the time of recurrence, it would be stage IV in most cases, so we omitted the stage. For the same reason, radiotherapy was also given to many patients at the time of initial treatment, but we considered the time of recurrence to be the starting point. In this study, we focused on long-term responders from the start of ICI administration.

  1. Was PD-L1 score affected the results?

Response: 165/246 (67%) were unable to measure PDL-1. For this reason, an overall survival study of 10 negative cases and 71 positive cases was performed, but there was no significant difference in survival. The point of this study was to examine the difference between the ongoing / toxicity / decision / PD groups, but as shown in Table 1, there was no difference in PDL-1 positivity/negativity between the groups.

  1. Was there any difference between recurrent tumor and primary carcinoma? 

Response:  In the present study, we did not consider differences between the primary tumor and the histological type, such as differentiation, in cases of recurrence.

  1. Was the primary site (nasopharynx, oropharynx and so on) not affect the results?

Response: There was actually no difference in survival rates by site for the 246 cases as a whole. It may be that site does not matter much once recurrence occurs. I did not include this survival curve directly in the manuscript. The purpose of my study was to answer the question: when should ICI be discontinued? Or should it not? was. From this point of view, as shown in table 1, there is no site-specific difference among the four groups, which indirectly shows that site is irrelevant.

Round 2

Reviewer 2 Report

Comments and Suggestions for Authors

This article aims to determine the optimal timing of discontinuation of immune checkpoint inhibitors in patients of head and neck squamous cell carcinoma. The aim of this study was very important. I agreed with the authors response to the comments.